# Autophagy: A Friend or Foe in Allergic Asthma?

**DOI:** 10.3390/ijms22126314

**Published:** 2021-06-12

**Authors:** Efthymia Theofani, Georgina Xanthou

**Affiliations:** 1Cellular Immunology Laboratory, Center for Basic Research, Biomedical Research Foundation of the Academy of Athens, 11547 Athens, Greece; etheofani@bioacademy.gr; 21st Department of Respiratory Medicine, “Sotiria” Regional Chest Diseases Hospital, Medical School, National Kapodistrian University of Athens, 11547 Athens, Greece

**Keywords:** autophagy, asthma, airway inflammation

## Abstract

Autophagy is a major self-degradative process through which cytoplasmic material, including damaged organelles and proteins, are delivered and degraded in the lysosome. Autophagy represents a dynamic recycling system that produces new building blocks and energy, essential for cellular renovation, physiology, and homeostasis. Principal autophagy triggers include starvation, pathogens, and stress. Autophagy plays also a pivotal role in immune response regulation, including immune cell differentiation, antigen presentation and the generation of T effector responses, the development of protective immunity against pathogens, and the coordination of immunometabolic signals. A plethora of studies propose that both impaired and overactive autophagic processes contribute to the pathogenesis of human disorders, including infections, cancer, atherosclerosis, autoimmune and neurodegenerative diseases. Autophagy has been also implicated in the development and progression of allergen-driven airway inflammation and remodeling. Here, we provide an overview of recent studies pertinent to the biology of autophagy and molecular pathways controlling its activation, we discuss autophagy-mediated beneficial and detrimental effects in animal models of allergic diseases and illuminate new advances on the role of autophagy in the pathogenesis of human asthma. We conclude contemplating the potential of targeting autophagy as a novel therapeutic approach for the management of allergic responses and linked asthmatic disease.

## 1. Mechanisms of Autophagy

Autophagy is a self-degradative process crucial for energy maintenance during development and response to stress. Autophagy can be selective and non-selective [1]. Non-selective autophagy is characterized by arbitrary engulfment and degradation of cytoplasmic material and occurs continuously at the basal level, facilitating the turnover and recycling of cytoplasmic contents. Under conditions of nutrient deprivation, autophagy is upregulated providing macromolecules essential for anabolic synthesis [1]. Selective autophagy is an evolutionarily conserved process mediated by the ubiquitination and subsequent degradation of specific subcellular targets, such as protein aggregates, microbes, dysfunctional and superfluous organelles, along with the generation of specific nutrients in response to environmental changes [2]. Selective autophagy also monitors lipid imbalance, glucose scarcity, amino acid deprivation, and iron shortage and facilitates metabolic reprogramming. Other types of selective autophagy include the capacity to clear intracellular pathogens (xenophagy), the degradation of damaged mitochondria (mitophagy), endoplasmic reticulum (ER-phagy), peroxisomes (pexophagy), polyubiquitinated aggregates (aggrephagy), ribosomes (ribophagy), lipid droplets (lipophagy), and cilia components (ciliophagy) [3].

Three main types of autophagy depend primarily on the delivery pathways of proteins and organelles to the lysosome: macroautophagy, microautophagy, and chaperone-mediated autophagy [4]. Macroautophagy hereafter referred to as autophagy, is triggered by a plethora of stimuli, including nutrient deprivation, oxidative stress, hypoxia, protein aggregates, and toxic molecules [5,6,7,8]. One of the best characterized autophagy-associated structures is the autophagosome, a double-membrane organelle. Autophagosome formation begins with the appearance of a membrane cup, called ‘phagophore’. The next step proceeds with the elongation of the autophagosome membrane which closes and forms the double-membranous autophagosome. Subsequently, the fusion of the autophagosome with a lysosome leads to the formation of the autolysosome that elicits the digestion of cellular components [5,6,7,8].

Autophagosome formation is regulated by a series of autophagy-related genes (*ATG*) [9,10]. The first step includes the formation of the ULK serine/threonine-protein kinase complex. Subsequently, the formation of the phagophore membrane is regulated by the class-III phosphatidylinositol 3-kinase (PtdIns3K) complex, constituted of Beclin-1, VPS34, VPS15, and ATG14L and localized in the endoplasmic reticulum [9,10]. The elongation and expansion of the phagophore membrane depend on two ubiquitin-like conjugation systems, including ATG5, ATG12, ATG16L1, ATG8/microtubule-associated protein 1 light chain 3 (LC3) recruited to the autophagosomal membrane by the PtdIns3K complex. The first system is associated with autophagosome assembly and the second with autophagosome formation and maturation [9,10,11]. The maturation of the autophagosome occurs when LC3-I is conjugated by the E1 enzyme ATG7 and the E2 enzyme ATG3 to phosphatidylethanolamine (PE) to form LC3-II. LC3 lipidation promotes its recruitment to autophagosomal membranes [8,9,10,11]. Ubiquitination is an important tag for adaptor proteins, including the selective autophagy receptor Sequestosome 1 (p62) that traffics cellular cargo to the autophagosome. Autophagosomes fuse with the lysosome once the vesicle membranes are closed and this is essential for the degradation of cellular substrates by lysosomal acid hydrolases and other degradative enzymes [8,9,10,11].

One of the best-characterized methods to monitor autophagy is the detection of autophagosome formation and other autophagy-related structures using electron microscopy and LC3 puncta evaluation by confocal microscopy. Western blot and immunofluorescent analyses of autophagosome/lysosome-related proteins, such as LC3, lamp1, ATG is also utilized. To monitor autophagic flux, it is essential to evaluate the conversion of LC3-I to LC3-II, concomitant with p62 degradation in cell lysates. Still, increased expression of autophagy-related proteins does not always indicate autophagy activation, as defective autophagosome fusion with the lysosomes may increase autophagy proteins, resulting in misleading conclusions. Hence, the use of appropriate controls including autophagy inducers (i.e., starvation or treatment with rapamycin) and inhibitors (i.e., bafilomycin A1, a potent V-ATPase inhibitor or chloroquine and hydroxychloroquine) is needed to correctly evaluate autophagy activation. A series of reporter and knockout mice for autophagy-related proteins are also available and represent powerful tools for monitoring autophagy activation and functions. Further information pertaining to the monitoring of autophagy is available and the reader is referred to other excellent papers [12,13].

Alterations in autophagy and/or mitophagy are associated with a diverse range of human diseases including metabolic disorders, neurological, cardiovascular, autoimmune and infectious diseases, and cancer [14,15,16,17,18]. Mutations in *PARKIN* and *PINK1* genes are associated with the pathogenesis of Parkinson’s disease [19,20,21], while the T300A mutation in the *ATG16L1* gene is associated with increased risk of Crohn’s disease [22,23,24]. Paget disease of the bone and amyotrophic lateral sclerosis are linked to mutations in the *P62* receptor [25]. Single nucleotide polymorphisms in the *ATG5* gene are associated with an enhanced risk of allergic asthma [26,27] and systemic lupus erythematosus [28,29]. Notably, dysregulations of the major autophagy controller, the transcription factor EB (TFEB), have been described in Parkinson’s, Huntington’s, and Alzheimer’s diseases and are associated with elevated intracellular protein aggregation and autophagy dysfunction [30,31,32,33,34,35]. Increased p62 accumulation, indicative of deficient autophagic flux, is also observed in human atherosclerotic plaques and participates in disease progression [36]. Impaired mitophagy is associated with mitochondrial dysfunction and increased reactive oxygen species (ROS) release that promote insulin resistance and diabetes pathogenesis [37]. Finally, lysosomal storage disorders, including mucopolysaccharidoses, mucolipidoses, oligosaccharidoses, Pompe disease, Gaucher disease, Fabry disease, Niemann-Pick disorders, and neuronal ceroid lipofuscinoses are characterized by lysosomal dysfunction and impaired autophagic flux [38,39].

Considering the key role autophagy plays in central biological processes and the pathophysiology of human diseases, several mechanisms have evolved to regulate deleterious effects elicited by overactive and/or impaired autophagy.

## 2. Autophagy Regulation

Activation of the metabolic sensor, mammalian target of rapamycin complex (mTORC) in response to environmental and intracellular stresses, represents a crucial process regulating autophagy [40]. MTORC1 signaling promotes cellular growth and proliferation through induction of anabolic pathways and inhibition of apoptosis and catabolic processes, including autophagy [8,40,41,42]. Specifically, mTORC1 induces inhibitory ULK1 phosphorylation and impedes autophagy activation. Under cellular stress, such as starvation conditions, mTORC1 activity is inhibited allowing ULK1 complex activation (Figure 1). MTORC1 also phosphorylates and inactivates regulatory subunits of the PIK3C3/VPS34 kinase complex and phosphorylates TFEB, retaining it in the cytoplasm or inducing its proteasomal degradation [43,44,45,46,47]. TFEB represents a master autophagy controller as, upon nuclear translocation, it regulates the transcription of genes involved in autophagy, mitophagy, and lipophagy [48,49,50]. Another metabolic regulator of autophagy is the AMP-activated protein kinase (AMPK) that senses changes in intracellular ATP/AMP concentrations [51]. Under starvation conditions, AMPK phosphorylates ULK1, initiating autophagy (Figure 1). Moreover, AMPK induces inhibitory phosphorylation of Raptor, the essential component of mTORC1 complex, and activating phosphorylation of the mTORC1 inhibitor, the tuberous sclerosis complex subunit 2 (TSC2) [52,53]. AMPK also phosphorylates subunits of the PIK3C3/VPS34 kinase complex [54,55]. Calmodulin-dependent protein kinase II (CaMKII), acting downstream of the Ca^2+^/calmodulin complex, phosphorylates Beclin-1 at Ser90 inducing its K63-linked ubiquitination and resulting in autophagy activation [56]. Free fatty acids (FFAs) activate the PIK3C3/VPS34 kinase complex through AMPK, MAPK8/JNK1, and EIF2AK2/PKR signaling pathways [57,58,59]. Beta-oxidation of FFAs generates acetyl CoA which feeds the TCA cycle and acts as a substrate for histone acetylation (Figure 1). When acetyl-CoA levels are reduced, there is a shift towards a chromatin low acetylation state which favors the deacetylation and transcription of pro-autophagic genes [60,61]. Low acetyl-CoA levels also increase NAD^+^ levels that activate the NAD^+^-dependent class III deacetylases, sirtuins (SIRTs), which enhance autophagy by deacetylating FOXO (forkhead box protein O) transcription factors and *ATG5* and *ATG7* loci [62,63] (Figure 1). Another central autophagy inducer is hypoxia. Hypoxia results in reduced energy charge due to low levels of ATP leading to AMPK activation and autophagy induction [64]. Moreover, hypoxia activates autophagy through enhanced production of ROS [65,66].

Apart from metabolic signals, immune mediators, including cytokines, are critical controllers of autophagic responses. Signaling downstream of the anti-inflammatory cytokine IL-10 and its receptor IL-10R inhibits starvation-induced autophagy in murine macrophages by activating the phosphatidylinositol 3-kinase (PI3K)/protein kinase B (AKT) signaling pathway [67]. In contrast, other studies have shown that IL-10 promotes autophagy in lipopolysaccharide (LPS)-stimulated bone marrow-derived macrophages and prevents the accumulation of dysfunctional mitochondria and ROS release through suppression of mTORC1 functions [68]. The Th1 cell-associated cytokine IFN-γ stimulates autophagy in macrophages [68,69]. In fact, in *Mycobacterium tuberculosis*-stimulated peripheral blood mononuclear cells (PBMCs), high IFN-γ production, correlates with increased LC3-II levels, proposing a protective role for IFN-γ-induced autophagy during bacterial infections [70]. In contrast, the Th2 cell-associated cytokines, IL-4 and IL-13 inhibit autophagy in murine and human macrophages, under starvation or IFN-γ stimulation [69]. However, IL-4 activates autophagy in CD4^+^ T cells, B cells, and dendritic cells (DCs), while IL-13 enhances autophagy and mucus secretion in airway epithelial cells, pointing to cell-type specific effects of these cytokines on autophagy activation [71,72,73]. IL-21 suppresses autophagy in activated CD4^+^ T cells, associated with defective differentiation and effector function [74]. Remarkably, autophagy induction in Th2 polarized cells prevents TCR activation through targeting Bcl10 for degradation and inhibiting NF-kB activation [75]. IL-2 upregulates LC3-II expression and autophagosome formation in mouse embryonic and primary lung fibroblasts, leading to enhanced proliferation and survival [76]. In lung epithelial cells, IL-17A stimulation inhibits BCL2 phosphorylation, preventing its degradation, and thus, allowing BCL2 and Beclin-1 interaction and autophagy attenuation [77]. IFN-γ enhances annexin A2 exosomal release by lung epithelial cells through autophagy (exophagy) induction [78].

Altogether, these studies highlight distinct and often opposing effects of autophagy on cell metabolism and effector functions that depend on the environmental stimulus, the context of the response, the cell-type and its activation status, and suggest that close monitoring of autophagic processes is essential for the protection against pathologic sequelae that drive human diseases.

## 3. Transcriptional Control of Autophagy

Autophagic responses are also controlled at the transcriptional level. Transcription factors documented to regulate autophagy gene expression, include TFEB, FOXO3, and p53 [79,80,81]. TFEB belongs to the microphthalmia/transcription factor E (MiT/TFE) family of transcription factors that include melanogenesis associated transcription factor (MITF), Transcription Factor EC (TFEC), and Transcription Factor Binding To IGHM Enhancer 3 (TFE3) proteins that are characterized by the recognition of coordinated lysosomal expression and regulation (CLEAR) motifs [82,83,84]. CLEAR motifs characterize the promoter region of lysosomal genes [33] (Figure 1). Studies in *Tfeb* and *Tfe3* knockout macrophages demonstrated that these transcription factors promote autophagy gene expression independently of each other, indicative of partially redundant functions [85]. Activation of TFEB drives its nuclear translocation wherein it promotes the transcription of autophagy and lysosome genes. Under nutrient-rich conditions, guanosine triphosphate (GTP)-bound heterodimeric RagGTPases (RagA) recruit mTORC1 to the lysosomal membrane, where it phosphorylates TFEB at serine/threonine residues, creating a binding site for the cytosolic chaperone-like protein 14-3-3 phospho-serine/phospho-threonine binding protein and resulting in TFEB cytosolic sequestration [47,84,86,87,88]. MTORC1 also impedes the nuclear localization of TFE3 and MITF [89]. ERK2 phosphorylates TFEB at Ser142, also inhibiting its nuclear translocation [88]. Under glucose starvation, AMPK activation promotes TFEB nuclear translocation, through the phosphorylation of ACSS2 (acetyl-CoA synthetase short-chain family member 2) [90] (Figure 1). ACSS2 binds to TFEB and initiates transcription of lysosome biosynthesis and autophagy genes. ACSS2 also generates acetyl-CoA that is used for histone H3 acetylation and autophagy gene induction [90]. The purinergic receptor P2 × 7 induces TFEB nuclear translocation through AMPK activation [91]. Notably, activation of the calcium and calmodulin-dependent serine/threonine phosphatase, calcineurin, dephosphorylates TFEB and triggers its nuclear translocation, pointing to a role for calcium signaling in autophagy induction [92]. Recent studies also demonstrated that Protein Kinase C Alpha (PRKCA) inhibits Glycogen Synthase Kinase 3 Beta (GSK3β) signaling, resulting in decreased TFEB phosphorylation and enhancement of its nuclear translocation [93].

FOXO3 represents a key transcriptional regulator of autophagy genes, including *ATG4*, *BECN1*, *LC3*, *ULK1*, and *VPS34* [80,94,95]. In response to growth factors and insulin stimulation, the activity of FOXO3 is inhibited by AKT, an upstream inducer of mTORC1, resulting in its cytoplasmic retention and attenuation of autophagy genes transcription (Figure 1). A recent study in *Caenorhabditis elegans* demonstrated that DAF16 (FOXO in mammals) cooperates with HLH30 (TFEB in mammals) to ensure appropriate expression of target genes during organismal responses to stressors, pointing to potential transactivating functions of FOXO and TFEB on autophagy gene induction [96].

P53, a tumor suppressor protein, inhibits mTORC1 lysosomal recruitment, through transcriptional induction of Sestrin proteins which activate AMPK [97,98]. Moreover, P53 induces the expression of a lysosomal protein called Damaged-regulated- modulator (DRAM) that enhances autophagy through an unknown mechanism [99]. P53 also controls the expression and activity of FOXO3 [100,101,102], and upon DNA damage, promotes TFEB/TFE3 nuclear translocation [103] (Figure 1). On the other hand, cytoplasmic p53 may act as a negative regulator of autophagy; however, the precise molecular mechanisms mediating this inhibitory effect remain elusive [81,104]. BCL2 Interacting Protein 3 (BNIP3) is another activator of autophagy, induced under hypoxia that disrupts the inhibitory binding of BCL-2 to Beclin1, promoting autophagosome biogenesis. BNIP3 is regulated by the transcription factors E2F1 and NF-kB; under normoxia, NF-kB binds to the *BNIP3* promoter, suppressing its expression. Under hypoxic conditions, reduced NF-kB binding to the *BNIP3* gene allows E2F1 binding to *BNIP3* regulatory elements promoting gene transcription [105]. E2F1 also controls the expression of other autophagy genes, such as *ULK1*, *LC3*, and *ATG5* [106].

Epigenetic modifications, including histone H3K9 dimethylation, H3K27 trimethylation, and H4K16 acetylation play an essential role in the regulation of autophagic responses [107] (Figure 1). H3K27 trimethylation, catalyzed by Enhancer of Zeste Homolog 2 (EZH2), suppresses the expression of negative regulators of mTORC1, leading to autophagy inhibition [108]. Bromodomain-containing protein 4 (BRD4) hinders autophagy and lysosomal gene transcription through the recruitment of the histone lysine methyltransferase G9a, which places a suppressive H3K9 dimethylation mark on their promoters [109]. In contrast, AMPK inhibits BRD4 activation, allowing lysosomal and autophagic gene induction. Notably, low amino acid or glucose concentrations enhance co-activator-associated histone arginine methyltransferase 1 (CARM1)-mediated dimethylation of H3 Arg17 on the promoters of autophagy and lysosomal genes, activating their transcription [110]. In fact, CARM1 exerts transcriptional co-activator function on autophagy-related and lysosomal genes through interactions with TFEB. Interestingly, recent studies have demonstrated that H4K16 acetylation controls autophagy gene expression associated with degradation of human Males absent On the First (hMOF/KAT8/MYST1), an H4K16 acetyltransferase [111]. More specifically, upon nutrient starvation, histone acetyltransferase hMOF/KAT8/MYST1 activity is reduced, leading to decreased acetylation of H4K16, which results in the transcriptional activation of autophagy genes. SIRT1-induced deacetylation of ATG proteins, such as ATG5, ATG7, and LC3, and the FOXO family of transcription factors are also involved in autophagy induction [112]. SIRT1 also deacetylates the Tumor suppressor serine/threonine-protein kinase LKB1 that activates AMPK signaling, increasing autophagy induction [113] (Figure 1). Finally, under nutrient-rich conditions, Forkhead box K (FOXK) drives the transcriptional repression of autophagy gene expression by binding to promoter regions of early-stage autophagy genes (e.g., ULK complex) and recruiting the SIN3A-Histone deacetylase (HDAC) repressor complex to these regions [114].

Collectively, the aforementioned paragraphs highlight the pleiotropic effects of metabolic, transcriptional, and epigenetic mechanisms on the regulation of autophagy and illuminate the existence of complex networks that are activated or inhibited in each cell to ensure cell survival and maintenance of its homeostasis during encounters with intracellular and extracellular stressors.

## 4. Allergic Asthma Immunopathogenesis

Asthma is a chronic heterogeneous lung disease characterized by airway hyperresponsiveness (AHR) to innocuous inhaled allergens and pulmonary inflammation [115]. Asthma encompasses complex and multiple clinical phenotypes that incorporate persistent symptoms and acute disease exacerbations [115,116,117]. Certain asthmatics exhibit severe asthma (SA) that is poorly controlled and represents a major health and socio-economic burden [116,117]. SA patients are characterized by frequent, and sometimes life-threatening, disease exacerbations and more severe symptoms, including shortness of breath, wheeze, cough, and increased mucus production [118,119]. These individuals require treatment with high-dose inhaled (or systemic) corticosteroids (CS) in combination with a second long-term (controller) medication and exhibit low lung function and persistent eosinophilia in the bronchoalveolar lavage fluid (BAL), along with high levels of neutrophils and exhaled nitric oxide [120]. Depending on the type of immune cells involved in disease pathogenesis, asthma endotypes are mainly categorized as type 2 asthma, characterized by Th2 cell-mediated and eosinophilic inflammation, and non-type 2 asthma, associated with Th1 and/or Th17 cell and neutrophilic inflammation [120,121,122]. Another type of asthma is the paucigranulocytic asthma (PGA) characterized by low-grade airway inflammation and low numbers of eosinophil or neutrophil numbers in the airways, compared to other asthma endotypes [123,124]. PGA is associated with airway smooth muscle (ASM) dysfunction, persistent airflow limitation, and AHR. The lack of an effective therapeutic regime for individuals with SA represents a serious clinical need without an obvious solution at present.

Inflammatory cells infiltrating the allergic airways, including DCs, eosinophils, neutrophils, mast cells, and lymphocytes, play a crucial role in the initiation and propagation of asthmatic responses (Appendix A). DCs in the lung mucosa take up allergens, reach the mediastinal lymph nodes, and present allergen components to naive T cells which differentiate into Th1, Th2, Th9, or Th17 cell subsets and initiate inflammatory responses upon relocalization to the airways [125]. The production of Th2 cell-associated cytokines, such as IL-4, IL-5, and IL-13, by Th2 cells enhances mucus production, bronchial remodeling, and the recruitment of innate effector cells, including mast cells, basophils, and eosinophils [126,127]. A group of asthmatics is characterized by increased neutrophilic inflammation, along with Th17 and Th1 cell infiltration that produce high levels of IFN-γ and IL-17 in the allergic airways that correlate with asthma severity [128]. Th9 cells and type 2 innate lymphoid cells (ILC2s) enhance the production of IL-2 by mast cells, leading to further expansion of ILC2s, which in turn activate Th9 cells in a positive feedback loop [129]. ILC2s are activated in response to alarmins released by airway epithelial cells, including IL-25, IL-33, and thymic stromal lymphopoietin (TSLP), and represent the first producers of Th2 cytokines that activate B cell, T cell, and granulocyte infiltration in the allergic lung (Appendix A) [129]. Apart from airway inflammation, SA is characterized by extensive airway remodeling and narrowing thickened epithelium, and fixed airflow obstruction (Appendix A) [130,131].

Emerging clinical, epidemiological, and experimental evidence has illuminated dysregulated autophagy as a principal mechanism underlying asthma pathogenesis with conflicting reports showing both detrimental and beneficial effects [131,132,133,134]. Hence, delineation of the precise role of autophagy in the regulation of asthmatic responses is essential for the restoration of lung homeostasis and the development of more effective therapeutic interventions. In the next section, we will delineate current knowledge on the role of autophagy in the attenuation or propagation of allergic airway inflammation and linked human asthma.

## 5. Role of Autophagy in Allergic Airway Inflammation In Vivo

The role of autophagy in allergen-driven inflammatory responses in the airways remains poorly understood. Studies using ovalbumin (OVA)-induced allergic airway inflammation (AAI) mouse models documented decreased expression of *Atg5*, *Lc3*, and *Beclin1* in lung homogenates and BAL macrophages, accompanied by reduced protein levels in OVA-treated mice, compared to PBS-control mice [135]. In contrast, other reports demonstrated increased expression of Lc3b by airway epithelial cells and elevated Atg5 levels in lung homogenates in a mouse model of cockroach-allergen induced AAI [136]. Certain studies report suppressive effects of autophagy associated with protection against AAI (Table 1). Indeed, treatment with simvastatin reduced airway inflammation and remodeling, attenuated AHR, and decreased the levels of IL-4, IL-5, IL-13, and IFN-γ in the BAL in OVA-challenged mice and this was associated with increased expression of autophagy proteins, such as Atg5, LC3B and Beclin1 in lung homogenates, as well as, enhanced autophagosome formation in the lung parenchyma [135] (Table 1). Notably, administration of the autophagy inhibitor, 3-Methyladenine (3-MA), reversed simvastatin-induced suppression of AAI and remodeling. Seminal studies in *Atg5* deficient mice demonstrated increased neutrophilic influx, AHR, airway inflammation, and goblet cell accumulation, concomitant with elevated IL-1β and IL-17A levels in whole lung lysates upon house dust mite (HDM) sensitization and challenge [137]. Using CD11c-specific *Atg5^−/−^* mice the authors further showed that impaired autophagy in DCs resulted in increased IL-1 and IL-23 release, spontaneous AHR, severe neutrophilic and Th17 cell-mediated airway inflammation, and glucocorticoid resistance, while adoptive transfer of *Atg5^−/−^* CD11c^+^ DCs aggravated lung inflammation and increased IL-17 release in the allergic airways (Table 2, Appendix A). Interestingly, using bone marrow chimeras, these studies revealed that autophagy deficiency in non-hematopoietic cells did not ameliorate allergic airway disease phenotype, excluding a detrimental role for autophagy activation in lung tissue-resident cells [137]. In agreement, in another mouse model of acute AAI induced by intravenous transfer of in vitro generated OVA-specific Th17 cells, treatment with rapamycin, an autophagy inducer, reduced pulmonary inflammation accompanied by increased recruitment of plasmacytoid dendritic cells and reduction of neutrophilic infiltration and CXCL-1 levels in the BAL [138] (Table 1, Appendix A). Remarkably, mice lacking *Atg7* in myeloid cells exhibited enhanced sterile lung inflammation, accompanied by submucosal thickening, goblet cell metaplasia, increased collagen content, heightened IL-1β, IL-18, and IL-17 levels in the lungs and serum and increased mortality following intraperitoneal LPS injection, while bleomycin administration aggravated pulmonary inflammation and induced severe fibrosis (Table 2, Appendix A) [139]. Similarly, other investigators using myeloid-specific *Atg7*-deficient mice demonstrated spontaneous pulmonary inflammation, elevated expression of *Tnfα*, *Il6*, *Ccl2*, *Cxcl1*, *Cxcl2* genes, and myeloid cell infiltration in the lung [140]. These mice also exhibited increased mitochondrial ROS production and heightened sensitivity of alveolar macrophages (AMs) to TLR4 ligands, including low doses of LPS. In a model of eosinophilic chronic rhinosinusitis (ECRS), mice with myeloid cell-specific deletion of *Atg7* exhibited aggravated eosinophilic and mast cell infiltration, mucosal thickening, and increased production of prostaglandin D2 [141]. Macrophage activation in *Atg7*^−/−^ mice was associated with increased IL-1β levels, while macrophage depletion or IL1β receptor blockade alleviated eosinophilic inflammation. Overall, these studies propose that baseline autophagy activation in myeloid cells contributes to the maintenance of lung homeostasis and the control of aberrant inflammatory AM responses.

Emerging evidence also highlights a prominent role for autophagy in airway epithelial cell responses. Using mice with an inducible epithelial cell-specific (under the Clara cell promoter) disruption of *Atg7*, recent studies documented that loss of autophagy in airway epithelial cells resulted in swelling of bronchial epithelial cells, accompanied by p62 accumulation and increased AHR to methacholine due to enhanced airway thickening [142] (Table 2, Appendix A). Mechanistically, bronchial epithelial cells from *Atg7* deficient mice exhibited enhanced expression of the cytoprotective genes *Nqo1*, *Txnrd1*, and activation of the Keap-Nrf2 pathway that was associated with p62 accumulation.

Contradictory data propose a detrimental role for autophagy in the initiation and progression of AAI. For example, in a mouse model of OVA-induced severe eosinophilic AAI, treatment with 3-MA or *Atg5* knockdown reduced LC3-II expression in lung-infiltrating eosinophils, attenuated AHR, and decreased inflammatory cell recruitment in the BAL and lung [143] (Table 2). Moreover, the levels of IL-5 but not IL-4, IL-13, and IFN-γ were decreased in the BAL, while in vivo anti-IL-5 administration reduced LC3II expression in allergic lungs. In agreement, other investigators showed that treatment with 3-MA right before the OVA challenge, attenuated pulmonary inflammation associated with decreased eosinophil extracellular trap formation [144] (Table 2, Appendix A). In fact, it was demonstrated that 3-MA treatment reduced eosinophil numbers, eosinophil peroxidase activity and goblet cell hyperplasia, extracellular DNA concentrations in the BAL, and dampened IL-5, IL-13, IFN-γ, TNF-α, IL-1β, nuclear factor kappaB (NFκB) p65 and ROS levels in the lung. Notably, luteolin administration inhibited OVA-induced airway inflammation, accompanied by decreased Beclin-1-PI2KC3 protein expression and enhanced PI3K/Akt/mTOR activation [145]. Interestingly, other studies demonstrated that knockdown of *Mtor* in airway epithelial cells resulted in increased recruitment of inflammatory cells and eosinophils in the BAL, enhanced mucus accumulation, exacerbated AHR and heightened IL-25 levels, following HDM and/or OVA challenge (Table 1, Appendix A) [146]. In support, the administration of EX-527, a SIRT1 inhibitor, suppressed airway inflammation and reduced IL-4, IL-13, and IFN-γ levels in the BAL, associated with enhanced mTOR activation and decreased autophagy induction, effects that were reversed by concomitant rapamycin treatment [147]. Further support of a pathogenic role for autophagy in AAI came from studies in *Atg16l1*-deficient mice which exhibited attenuated mucus secretion upon intranasal IL-33 administration [75]. In support, 3-MA administration in a mouse model of cockroach allergen-induced AAI dampened lung inflammation, mucus production, AHR, ROS release, and Th2 cell-associated cytokines [136], (Table 2). Interestingly, miR-135-5p negatively regulated p62 expression and decreased inflammatory cytokine and chemokine release in a rat basophil cell line [148], highlighting miR-mediated antagonism as a novel mechanism of autophagy regulation. *Lc3-b* deficient mice were also characterized by reduced allergen-induced airway inflammation and mucus production [146]. Still, AHR was enhanced in *Lc3-b* deficient mice, pointing to differential effects of autophagy on AHR and pulmonary inflammation (Table 2).

Autophagy also controls pro-allergic responses elicited by innate immune cells. Pioneering studies revealed that *Atg5* deficiency in ILC2s decreased Th2 cytokine release, impaired fatty acid oxidation, and induced the accumulation of dysfunctional mitochondria [149] (Table 2, Appendix A). Remarkably, adoptive transfer of *Atg5^−/−^* ILC2 cells attenuated IL-33-induced pulmonary inflammation and AHR, while autophagy activation in ILC2-specific *Tfeb^T^*^G^ mice enhanced Th2 cytokine release, highlighting a central role for autophagy in the effector function and metabolic responses of ILC2s. B cell responses are also controlled by autophagy in the context of AAI. More specifically, mice lacking *Atg5* in B cells exhibit lower levels of IL-4, IL-13, and inflammatory cell numbers in the BAL, decreased OVA-specific IgE production in the serum, attenuated mucus production and antigen-presenting functions, concomitant with increased apoptosis and glycolysis [72] (Table 2, Appendix A). Notably, IL-4 induced JAK signaling was essential for autophagy activation in B cells.

In a chronic asthma model, deficient autophagic flux and increased p62 expression were detected in lung homogenates and were associated with exacerbated airway remodeling [150]. Interestingly, a recent study using a mouse model of HDM-induced acute and chronic AAI revealed that autophagy inhibition via intranasal administration of chloroquine (CQ) decreased inflammatory cell infiltration and TGF-β production in the BAL and attenuated AHR [151]. Moreover, CQ administration decreased collagen and mucus deposition, reduced airway remodeling, and lowered BECLIN-1 and ATG-5 protein levels in the allergic lungs (Table 2, Appendix A). A detrimental role for autophagy in airway remodeling was also noted in studies involving oral administration of astragalin, a kaempferol-3-O-glucoside, in OVA-sensitized mice (Table 2, Appendix A) [152]. Indeed, the investigators reported that astragalin treatment prevented the subepithelial deposition of collagen fibers and was associated with reduced beclin-1 and LC3 expression in airway epithelial cells and lung tissue. Remarkably, administration of JTE-013, a Sphingosine-1-phosphate receptor 2 (S1PR2) antagonist, in a mouse model of OVA-induced chronic airway inflammation and remodeling, decreased inflammatory cell recruitment and mucus production, reduced BAL IL-1, IL-4, IL-5, and serum IgE levels and attenuated collagen deposition and smooth muscle cell-activating proteins in the lungs. JTE-013 effects were accompanied by decreased Beclin-1 levels and LC3II/LC3I conversion and increased p62 accumulation, indicative of dampened autophagy [153].

Autophagy is critically involved in host defense mechanisms against invading pathogens, including respiratory tract infections. In fact, bacteria and viruses attack autophagy (xenophagy) early on the following infection, to avoid autophagosome formation and linked destruction [154]. In this respect, *Atg7* deficiency following *Pseudomonas aeruginosa* infection impaired pathogen clearance increased neutrophilic inflammation, and resulted in elevated IL-1β production [155]. Furthermore, in a mouse model of Human Rhinovirus 1B (HRV1B) infection, the anti-viral effects of budesonide administration were associated with autophagy activation [156]. In sharp contrast, baf-A1-induced autophagy blockade impeded influenza A virus replication in lung epithelial cells [157], while autophagy induction was essential for the formation of double-membrane vesicle-bound major histocompatibility complex (MHC) replication complexes in a mouse model of hepatitis virus infection [158]. Notably, respiratory syncytial virus (RSV)-infected *Map1lceb^−/−^* mice exhibited aggravated IL-17A-dependent lung pathology, while *Becn1^+/−^* mice displayed decreased Th2 cytokine release, mucus secretion, and lung inflammation, illuminating important tissue-targeting effects of autophagy in the context of certain viral infections [159]. Still, further mechanistic studies are needed to explore the precise role of autophagy in respiratory tract infections and pathogen-induced asthma exacerbations.

## 6. Activation of Autophagy in Human Asthma

Early studies discovered genetic polymorphisms of the *ATG5* and *ATG7* genes in individuals with pediatric and adult asthma that were linked to airway remodeling and impairment in respiratory system mechanics [23,24,160]. Still, subsequent studies did not detect an association of *ATG5* gene polymorphisms with asthma severity but only with higher sputum neutrophil numbers [161]. Increased autophagosome formation was observed in fibroblasts and airway epithelial cells, and heightened expression of LC3B and ATG5 was detected in lung biopsies from asthmatic patients compared to healthy controls [24]. Other studies did not report activation of autophagy in large airway epithelial cells in asthmatics, as monitored by ATG5, Beclin-1, and p62 expression in lung sections, while enhanced ATG5 and Beclin-1 levels, accompanied by reduced p62 expression, were detected in large ASMs [151,160]. In sharp contrast, recent studies showed that ATG5 expression in airway epithelium, ASM, and inflammatory cells was not increased in asthmatics and did not correlate with asthma severity or lung function [161]. These contradictory findings may be due to the distinct methodological approaches utilized, as well as, due to differences in patient cohorts or the timing of the analyses. Hence, further studies are needed to delineate the activation patterns of autophagy in the asthmatic lung and its potential correlations with disease onset and/or severity.

Interestingly, a protective role was identified for the complement regulatory protein CD46 in nasal airway epithelial cells from asthmatic individuals through enhancing autophagy [162] (Appendix A). More specifically, it was demonstrated that airway epithelial cells from asthmatics exposed in vitro to *Dermatophagoides pteronyssinus* exhibited increased autophagosome formation, LC3II expression, decreased apoptosis, and lower pro-IL-1β and NLRP3 levels following CD46 mAb crosslinking, while treatment with 3-MA reversed the anti-inflammatory effects of the CD46 mAb on these cells. Interestingly, upon exposure of human bronchial epithelial cells (BECs) to particulate matter, autophagy induction enhanced mucus secretion, protecting these particles [163]. In contrast, *ATG5* and *ATG14* knockdown in primary human tracheal epithelial cells stimulated with IL-13 decreased MUC5AC secretion and CCL26 (eotaxin-3) and ROS release, suggesting that autophagy activation in this setting is pathogenic [73] (Appendix A). Furthermore, in vitro stimulation of human BECs with *Alternaria* extract enhanced LC3 conversion and p62 degradation associated with elevated IL-18 levels [164]. Notably, treatment with 3-MA and Baf-A1, suppressed IL-18 release by *Altenaria*-stimulated BECs, highlighting an autophagy-induced unconventional mechanism of IL-18 secretion (Appendix A). In other studies, IL-1β increased LC3-II expression and IL-8 production by human small airway epithelial cells, while 3-MA, CQ, the PI3K inhibitor LY294002 or knockdown of ATG5 and Beclin-1 reversed IL-1β effects [165]. IL-13 or IL-33 treatment also enhanced autophagy in human airway epithelial cells (HAECs) from asthmatics, concomitant with a decrease in mTOR activity [146]. Remarkably, a protective role was proposed for autophagy in preventing ferroptotic cell death induced by Th2 inflammatory conditions on HAECs from asthmatics, while increased LC3II levels in HAECs correlated with lower mitochondrial DNA in the BAL fluid [166]. Notably, cockroach allergen-stimulated HAECs displayed decreased mitochondrial ROS release upon 3-MA treatment, while mechanistic studies demonstrated that oxidized CaMKII activation was essential for cockroach allergen-mediated autophagy activation [136].

Pertinent to human inflammatory cells, Ban et al. (2015) detected elevated LC3 puncta formation in the cytoplasm of sputum granulocytes and peripheral blood cells in patients with SA, compared to non-severe asthma patients and healthy controls [167]. More specifically, peripheral blood eosinophils and other PBMCs from SA patients exhibited higher autophagy levels, as evidenced by increased LC3B expression at baseline, which were further upregulated upon IL-5 or inhibited by 3-MA treatment. Other studies showed that peripheral blood neutrophils (PBNs) from individuals with SA had elevated LC3-II levels and NET production both at baseline and following ex vivo IL-8 stimulation [168] (Appendix AIncreased NET production by PBNs correlated with autophagy, while IL-8-induced NET formation negatively correlated with lung function measurement, such as FEV1/FVC. Moreover, treatment with CQ decreased PBNs migration towards IL-8 (Appendix A). Still, *ATG5* knockout did not compromise the ability of neutrophils or eosinophils to form extracellular traps [169]. Notably, activation of autophagy has been shown to elicit caspase-independent cell death in eosinophils and neutrophils under inflammatory conditions, emphasizing potential protective effects of autophagy against the presence of activated granulocytes [169,170].

Human bronchial fibroblasts (HBF) from individuals with SA exhibit enhanced protein expression of LC3-ΙΙ, LAMP2A, Pink1, and Sirt1 and mitochondrial damage, compared to cells from healthy volunteers [171]. Moreover, in vitro stimulation of HBF from asthmatics with IL-17 further increased mitochondrial dysfunction, collagen and fibronectin mRNA levels, and expression of LC3-ΙΙ and p62. Inhibition of autophagy using bafilomycin-A1 restored IL-17-mediated increase in PINK1 protein levels in HBFs from asthmatics and decreased pro-fibrotic gene expression, illuminating a role for IL-17-induced autophagy in promoting HBF fibrotic responses (Appendix A). Additionally, HBF from asthmatics exhibited enhanced mitochondrial depolarization and increased mRNA expression of Pink/Parkin pro-fibrotic and pro-inflammatory cytokines, such as IL-6, IL-8, COL1a1, COL3A1, and FN1, while treatment with 3-MA decreased pro-fibrotic and pro-inflammatory gene expression [172] (Appendix A). Interestingly, in vitro stimulation of human ASM cells with TGF-β1 increased Beclin-1 and LC3II levels, concomitant with enhanced collagen I expression, while the addition of CQ reversed TGF-β1 effects, pointing to pro-fibrotic effects of autophagy activation in human ASM cells [151] (Appendix A). In contrast, a protective role of autophagy was detected in bronchial smooth muscle cells from asthmatics, as evidenced by enhanced survival, concomitant with decreased Th2 cytokine release in simvastatin-treated cells co-administered with rapamycin (Appendix A) [135]. In support, 3-MA treatment reversed simvastatin-induced anti-inflammatory effects on human bronchial smooth muscle cells. Further studies showed that knockdown of p62 inhibited the in vitro proliferation and migration of bronchial smooth muscle cells from asthmatics, associated with decreased glucose consumption and lactate production, whereas p62 overexpression had the opposite effects, highlighting a cross-regulation between autophagy and metabolic responses in human ASM cells [150] (Appendix A).

The increase in autophagosome formation and p62 levels in allergic airways reported in certain studies could be due either to enhanced autophagic flux and autophagy activation or to inhibition of autophagosome-lysosome fusion, indicating impaired autophagy. As such, studies showing increased expression of autophagy proteins in lung-resident and inflammatory immune cells in asthmatics should be carefully evaluated and accompanied by detailed time course, functional and mechanistic analyses to delineate the precise role of autophagy in these cells.

## 7. Concluding Remarks and Future Perspectives

Autophagy displays both protective and detrimental roles in allergic airway inflammation and linked asthma depending on the cell type, the lung micromilieu, and the cell-intrinsic bioenergetic requirements. In fact, autophagy seems to play a balancing role intended to avoid excessive lung tissue damage, while ensuring a protective anti-pathogen response. For example, at baseline, activation of autophagy maintains homeostasis of lung-resident cells, while during respiratory tract infections, autophagy induction in infiltrating immune cells, such as macrophages and DCs, is essential for the elimination of pathogens and the activation of pathogen-specific T effector responses. Still, impaired autophagy or persistent autophagy activation during the chronic phase of the allergic response can enhance autophagosome accumulation and activate lung-infiltrating innate immune cells, as well as, airway epithelial cells, leading to a decline in lung function. Autophagy also represents a critical regulator of fibrosis and can enhance extracellular matrix (ECM) production (e.g., collagen, fibronectin) in ASM and mesenchymal cells, leading to airway wall thickening and rigidity [151,171]. Conversely, autophagy has been shown to mitigate fibrosis through its involvement in the degradation of ECM proteins [173]. In this respect, the documented genetic correlation of polymorphisms in *ATG* genes with asthma may be an inherent protective mechanism to reduce chronic airway inflammation and remodeling and therefore, correlations or associations of autophagy genes with lung function changes should not be associated with causal mechanisms [23,24,160].

Apart from autophagy, the role of mitophagy in allergic responses remains poorly defined and should be further explored. Mitochondrial depolarization, along with elevated ROS production, are associated with atopy, atopic dermatitis, and asthma [174,175,176,177]. Dysregulated mitochondrial bioenergetics may weaken airway epithelial cell integrity, enhance their fragility and lead to impaired secretion [178]. Considering that mitophagy may be essential for the restoration of allergen-induced mitochondrial dysfunction and linked phenotypic changes in asthma, targeting mitophagy may possess therapeutic potential. For example, rapamycin and metformin, as general autophagy-inducing drugs, preserve energy metabolism through regulating mitophagy and mitochondrial biogenesis stimulation [179,180]. Moreover, naturally occurring compounds, such as spermidine, resveratrol, and urolithin A, enhance mitochondrial integrity through mitophagy activation and exert potent anti-inflammatory effects [181]. Investigation of the role of these mitophagy-inducing substances in the protection against mitochondrial dysfunction and linked AAI is timely and needed.

Several questions regarding the precise role of autophagy in the pathophysiology of human asthma remain opaque. Considering that autophagy represents a versatile immune modulator, a better understanding of the interplay between autophagy and immune responses in the allergic airways is needed and expected to have important therapeutic applications for asthma. Still, targeting autophagy therapeutically, using autophagy activators or inhibitors, when the autophagic response is different in distinct cellular compartments of the lung is challenging. Additionally, although selective autophagy substrates have been identified, the physiological significance of degradation of each substrate in lung-resident and airway-infiltrating immune cells needs to be further explored. Critical questions also relate to the successful identification of autophagy biomarkers that determine autophagic activity in allergic diseases, particularly when monitoring drug effectiveness. Moreover, currently available autophagy-modulating drugs are not specific and the development of more specific autophagy activators and inhibitors is required before their use in preclinical and clinical studies. Future studies should also consider airway-targeted autophagy regulators that can be administered locally, such as, the development of a novel inhaler, or nanoparticle-based cell-targeted methods to avoid non-specific potentially detrimental systemic side effects. Interestingly, administration of certain autophagy inducers, such as trehalose, has been shown to inhibit cytomegalovirus infection, suggesting that autophagy could be also exploited as a therapeutic regime towards virus-induced asthma exacerbations [132]. Considering that currently utilized asthma immunotherapies, including montelukast, anti-IL-5, and anti-IgE antibody, can inhibit autophagy, their potentially detrimental effects on cell types wherein autophagy is protective should be also carefully evaluated [178]. Finally, the specific asthma endotype and stage of the disease should be taken under consideration when targeting autophagy systemically or in specific cell types.

In conclusion, studies using both animal models and clinical samples from asthmatic individuals, along with advanced transcriptomics, proteomics, and metabolomic analyses at the single-cell level, are imperative to delineate the precise mechanisms underlying autophagy activation and inhibition, particularly after exposure to environmental pollutants and allergens, and will ultimately help identify appropriate therapeutic targets that can effectively control severe treatment-refractory asthmatic responses.

## Figures and Tables

**Figure 1 ijms-22-06314-f001:**
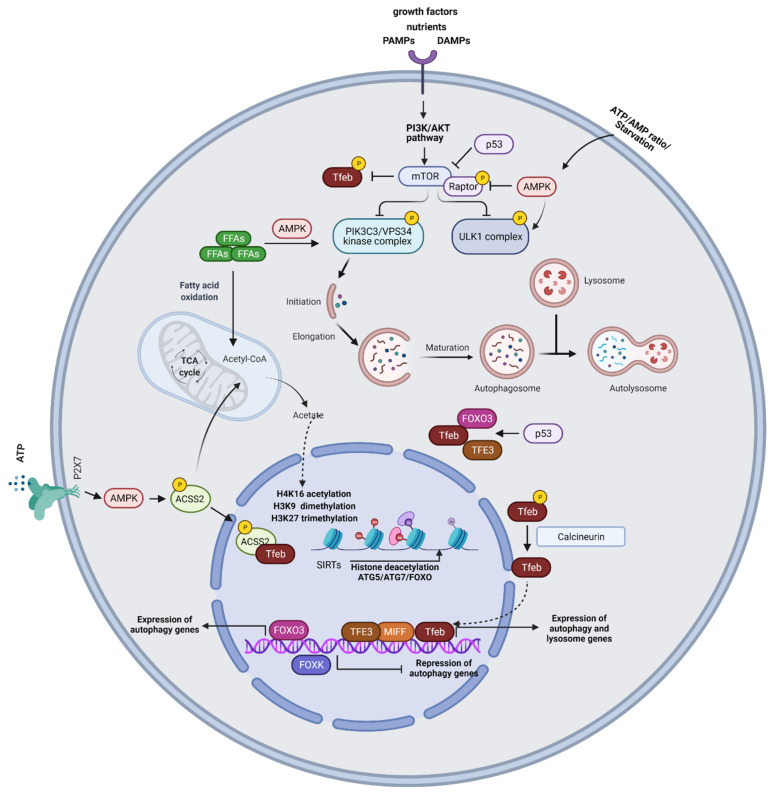
Mechanisms of autophagy regulation. MTORC1 induces inhibitory ULK1 phosphorylation and impedes autophagy activation. Under cellular stress, such as starvation conditions, mTORC1 activity is inhibited allowing ULK1 complex activation. MTORC1 also phosphorylates and inactivates regulatory subunits of the PIK3C3/VPS34 kinase complex and phosphorylates TFEB. AMPK senses changes in intracellular ATP/AMP concentrations and under starvation conditions, AMPK phosphorylates ULK1, initiating autophagy. Moreover, AMPK induces inhibitory phosphorylation of Raptor. Free fatty acids (FFAs) activate the PIK3C3/VPS34 kinase complex through AMPK. Beta-oxidation of FFAs generates acetyl CoA which feeds the TCA cycle and acts as a substrate for histone acetylation. SIRTs enhance autophagy by deacetylating FOXO, *ATG5*, and *ATG7*. Activation of TFEB drives its nuclear translocation wherein it promotes the transcription of autophagy and lysosome genes. Under glucose starvation, AMPK activation promotes TFEB nuclear translocation, through the phosphorylation of ACSS2. ACSS2 binds to TFEB and initiates transcription of lysosome biosynthesis and autophagy genes. ACSS2 also generates acetyl-CoA that is used for histone H3 acetylation and autophagy gene induction. The purinergic receptor P2X7 induces TFEB nuclear translocation through AMPK activation. FOXO3 represents a key transcriptional regulator of autophagy genes. P53 controls the expression and activity of FOXO3 and promotes TFEB/TFE3 nuclear translocation. Epigenetic modifications, including histone H3K9 dimethylation, H3K27 trimethylation, and H4K16 acetylation play an essential role in the regulation of autophagic responses. SIRT1-induced deacetylation of ATG proteins, such as ATG5, ATG7, and LC3, and the FOXO family of transcription factors are also involved in autophagy induction.

**Table 1 ijms-22-06314-t001:** Activation of autophagy and its effect on Allergic Airway Disease Outcome.

Autophagy Activation	
Treatment	Allergic Airway Disease Outcome
*Mtor*^−/−^ bronchial epithelial cells in HDM and/or OVA challenged mice	Increased recruitment of inflammatory cells and eosinophils in the BAL Enhanced mucus accumulation Exacerbated AHR Heightened IL-25 levels
ILC2-specific *Tfeb^T^*^G^ mice in IL-33-induced AAI	Enhanced ILC2s cell infiltration in the lungs Increased survival and proliferation of ILC2s Enhanced Th2 cytokine release
Simvastatin administration in OVA-challenged mice	Reduced airway inflammation and remodeling through autophagy activation Attenuated AHR Decreased IL-4, IL-5, IL-13, and IFN-γ levels in the BAL
Rapamycin administration in acute AAI induced by intravenous transfer of in vitro generated OVA-specific Th17 cells	Reduced pulmonary inflammation Increased recruitment of plasmacytoid dendritic cells Reduction of neutrophilic infiltration in the BAL Reduced CXCL-1 levels in the BAL

**Table 2 ijms-22-06314-t002:** Deficiency of autophagy and its impact on Allergic Airway Disease Outcome.

Autophagy Deficiency	
Treatment	Allergic Airway Disease Outcome
CD11c-specific *Atg5^−/−^* mice (HDM mouse model)	Increased IL-1β and IL-23 release Increased AHR Severe neutrophilic and Th17 cell-mediated airway inflammation Glucocorticoid resistance
Myeloid specific Atg7^−/−^ mice (LPS or bleomycin)	Increased IL-1β, IL-18, and IL-17 levels in the lungs and serum Increased mortality
Atg7^−/−^ airway epithelial-specific mice	Swelling of bronchial epithelial cells Increased AHR Enhanced airway wall thickening Increased p62 accumulation
*Atg16l1*^−/−^ mice (intranasal IL-33 administration)	Attenuated mucus secretion
3-MA or Atg5^−/−^ mice (OVA-induced severe eosinophilic AAI)	Attenuated AHR Decreased inflammatory cell recruitment in the BAL and lung Reduced IL-5 levels in the BAL
Lc3-b^−/−^ mice (HDM and/or OVA AAI)	Reduced airway inflammation and mucus production Increased AHR
Atg5^−/−^ ILC2s specific mice (IL-33-AAI)	Decreased Th2 cytokine release Impaired fatty acid oxidation Attenuated pulmonary inflammation and AHR
Atg5^−/−^ B cell-specific mice (OVA-induced AAI)	Reduced levels of IL-4, IL-13, and inflammatory cell numbers in the BAL Decreased OVA-specific IgE production Reduced mucus production
3-MA administration (mouse model of cockroach allergen-induced AAI)	Decreased lung inflammation and mucus production Attenuated AHR Reduced ROS release Decreased Th2 cell-associated cytokines
3-MA administration (OVA-induced AAI)	Attenuated pulmonary inflammation Reduced eosinophil numbers, eosinophil peroxidase activity, and extracellular DNA concentrations in the BAL Reduced ROS levels Decreased goblet cell hyperplasia
CQ administration (HDM-induced acute and chronic AAI)	Decreased inflammatory cell infiltration Reduced TGF-β production in the BAL Attenuated AHR Decreased collagen deposition and mucus production Reduced airway remodeling
Astraglin administration (OVA-induced AAI)	Decreased the subepithelial deposition of collagen fibers through autophagy inhibition
Luteolin administration (OVA-induced AAI)	Decreased inflammatory cell infiltration Reduced IL-4, IL-5, IL-13 levels in the BAL Decreased collagen deposition and mucus production
EX-527 administration (OVA-induced AAI)	Decreased airway inflammation Reduced IL-4, IL-13, and IFN-γ levels in the BAL
JTE-013 administration (OVA-induced chronic AAI)	Decreased inflammatory cell recruitment Reduced IL-1, IL-4, IL-5 levels in BAL Reduced mucus production Attenuated collagen deposition and smooth muscle cell activation

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
