# Peer review of "Autophagy: A Friend or Foe in Allergic Asthma?"

_ijms, 2021, doi:10.3390/ijms22126314_

Round 1

Reviewer 1 Report

To Author:

The provided manuscript is well written and organized. The complex and dynamic role of autophagy in asthma is clearly presented in the manuscript. The manuscript is suitable for publication but may benefit from a few minor changes.

Minor Suggestions

  • A figure delineating the mechanism, at the protein/organelle level, described in the “Autophagy regulation” section would significantly improve the manuscript. The reader would benefit from such a figure and make the cell-by-cell molecular mechanisms described in later sections more digestible.
  • The figure presented in the manuscript has little impact and might be better suited to be converted to a table. The cell specific details are important and should be included in some form (i.e. a table).
  • Lines 202 and 205 appear to have typos/spelling errors, please check.
  • The sentence from line 239 to 241 is overly complex and runs on, please simplify.

Author Response

Author response:

We are grateful to the Reviewer for the positive and insightful comments that have considerably improved the quality of our review.

The following changes have been made to the manuscript:

  1. Following the Reviewer’s suggestions, we have designed and included a figure (main Figure 1) that delineates the mechanisms that regulate autophagy at the intracellular and intranuclear levels.
  2. We have moved the original Figure 1 to the supplement (now shown as Supplementary Figure 1) and included some additional studies to the Tables, as suggested by both Reviewers.
  3. We have corrected the typos/spelling errors on lines 202 and 205.
  4. We have simplified the sentence on lines 239-241 and divided it into two separate sentences.

Reviewer 2 Report

The review “Autophagy: a friend or foe in allergic asthma? “ by  Efthymia Theofani1 and Georgina Xanthou is an interesting paper focusing on a  key process involved in immune responses, inflammation, and antiviral immunity, which  opens a new field in allergy and asthma research.

The problem I have with this review is that only the second part is actually focused on the topic.

I suggest Authors to shorten the introductory part, as excellent reviews on autophagy are available, among them the review by Saori R. Yoshii  and Noboru Mizushima, which appeared  in this Journal in 2017.

Additional points.

1- A brief  section on methods used to analyze autophagy should be added,   as autophagy involves dynamic and complicated processes and  it is often incorrectly investigated.

2- The observation of Kielan D McAlindene al. (AmJ Respir Cell Mol Biol 2019) who demonstrate cell context-dependent and selective activation of autophagy in structural cells in asthma should be reported.

3- Concerning therapeutic intervention targeting autophagy, Authors should add more recent papers on this topic:

-Wang S et al. Luteolin inhibits autophagy in allergic asthma by activating PI3K/Akt/mTOR signaling and inhibiting Beclin-1-PI3KC3 complex. Int Immunopharmacol 2021

-Li W et al. MTOR suppresses autophagy-mediated production of IL25 in allergic airway inflammation. Thorax 2020

-Wu Y et al. Suppression of sirtuin 1 alleviates airway inflammation through mTOR‑mediated autophagy. Mol Med Rep. 2020 Sep;22(3):2219-2226.

-Liu H et al. S1PR2 Inhibition Attenuates Allergic Asthma Possibly by Regulating Autophagy. Front Pharmacol. 2021

4- References 81 and 84 are identical

Author Response

Author response:

We are thankful to the Reviewer for the valuable comments and suggestions. We have incorporated the comments of the Reviewer and believe the manuscript has been substantially improved.

The following changes have been made to the manuscript:

  1. We have significantly reduced the parts of the review describing the mechanisms involved in autophagy regulation.
  2. We have included a short section on the methods utilized to analyze autophagy, as suggested by the Reviewer.
  3. The findings of Kielan D McAlindene al. (AmJ Respir Cell Mol Biol 2019) had been included in the original manuscript but we have emphasized them now in the text.
  4. We have added to the manuscript the references suggested by the Reviewer pertinent to therapeutic intervention of allergic airway inflammation targeting autophagy.
  5. We have corrected references 81 and 84.
